# Spatio-Temporal Distribution Characteristics of Glacial Lakes in the Altai Mountains with Climate Change from 2000 to 2020

**Nan Wang [1], Tao Zhong [1], Jianghua Zheng [1,2,\*], Chengfeng Meng [1] and Zexuan Liu [1]**

[1] College of Geography and Remote Sensing Sciences, Xinjiang University, Urumqi 830017, China; wsf@stu.xju.edu.cn (N.W.)

[2] Key Laboratory of Oasis Ecology, Xinjiang University, Urumqi 830017, China

\* Correspondence: zheng.jianghua@xju.edu.cn

**Abstract:** The evolution of a glacial lake is a true reflection of glacial and climatic change. Currently, the study of glacial lakes in the Altai Mountains is mainly concerned with the application of high-resolution remote sensing images to monitor and evaluate the potential hazards of glacial lakes. At present, there is no rapid and large-scale method to monitor the dynamical variation in glacial lakes in the Altai Mountains, and there is little research on predicting its future tendency. Based on the supervised classification results obtained by Google Earth Engine (GEE), combined with an analysis of meteorological data, we analyzed the spatial and temporal variations in glacial lakes in the Altai Mountains between 2000 and 2020, and used the MCE-CA-Markov model to predict their changes in the future. According to the results, as of 2020, there are 3824 glacial lakes in the Altai Mountains, with an area of 682.38 km$^2$. Over the entire period, the glacial lake quantity growth rates and area were 47.82% and 17.07%, respectively. The distribution of glacial lakes in this region showed a larger concentration in the north than in the south. Most glacial lakes had areas smaller than 0.1 km$^2$, and there was minimal change observed in glacial lakes larger than 0.2 km$^2$. Analyzing the regional elevation in 100 m intervals, the study found that glacial lakes were predominantly distributed at elevations from 2000 m to 3000 m. Interannual rainfall and temperature fluctuations in the Altai Mountains have slowed since 2014, and the trends for the area and number of glacial lakes have stabilized. The growth of glacial lakes in both number and surface area is expected to continue through 2025 and 2030, although the pace of change will slow. In the context of small increases in precipitation and large increases in temperature, in the future, glacial lakes with faster surface area growth rates will be located primarily in the southern Altai Mountains.

**Keywords:** Altai Mountains; glacial lakes; GEE; climate change; driving force analysis; simulation and prediction



## 1. Introduction

Glacial lakes act as terrestrial storage areas for glacial meltwater [1], and are susceptible to climate change [2]. In recent years, global temperatures have been rising, causing glacial lakes to grow rapidly with climate change and glacial retreat. These glacial lakes have the potential to serve as valuable water resources in alpine regions [3]; however, if these lakes burst, there are potential risks to downstream populations and infrastructure [4]. As of August 2021, according to Working Group I of the IPCC's Sixth Assessment Report (AR6), the Earth's surface was projected to warm by approximately 1.5 °C to 1.6 °C within the next two decades [5]. Against the background of global warming, it is therefore of great importance to investigate and analyze the response relationship between glacial lakes and climate change, to establish disaster warning systems, and exploit the glacial water resources.

At present, the focus of research on glacial lakes mainly includes methods for extracting glacial lake information based on remote sensing, the analysis of glacial lake temporal

and spatial characteristics and influencing factors, and the identification of potentially dangerous glacial lakes. Using the methods of water index data extraction and artificial visual interpretation, Wang et al. extracted the attribute information for glaciers and glacial lakes in the Hengduan Mountains between 1990 and 2015 [6]. On the basis of GEE, Shugar and his colleagues used remote sensing images from 1990 to 2015 to build a spatial distribution model and the evolutionary features of global glacial lakes [1]. Jain studied glacial lakes in the western Himalayas using high-resolution Google Earth images and Landsat images, and proposed that the main sources of energy fluctuation in glacial lakes were solar radiation, temperature, and precipitation [7]. Zheng et al. used water classification algorithms and visual interpretation to outline glacial lake boundaries on Landsat images, to comprehensively assess their historical development and current status, and simulated future changes in glacial lakes of the Third Pole region, along with the related glacial lake outburst flood (GLOF) risk [8]. There are an increasing number of studies on dynamic change monitoring for glacial lakes, but most have been concentrated in the Himalayas [7–11]. The Central Asian region is likely to become warmer and wetter in the context of global warming [12], which will influence the shrinkage of glaciers and the extension of glacial lakes. The former research indicated that the acceleration of glacial melting caused by global warming is the main reason for the growth of glacial lakes in the Kunlun Mountains [13,14]. From 1990 to 2015, the number and area of glacial lakes in the eastern Tianshan Mountains increased by 64.06% and 47.92%, respectively. The rapid warming of the climate, and the decline in glaciers are the major causes for the accelerated expansion of glacial lakes in the region [15].

Over the past few years, glaciers in the Altai Mountains have shown a trend of retreat. Between 1990 and 2011, the volume of glacial meltwater in the region was four times that of the Irtysh River's mean yearly flow rate [16]. The observation of glacial lake changes in the Altai Mountains can be used as a basis for developing and utilizing water resources in arid areas, and also for evaluating the hazard of glacial lake outburst disasters. Klinge studied glacial and lake sediments in the Altai region of Mongolia, and noted that lower temperatures caused reduced evaporation rates, and were the primary triggers for elevated lake levels during glacial advances [17], but did not analyze all glacial lakes in the Altai Mountains. Luo studied the dynamics of lakes in the Altai Mountains, and their reaction to climate change from 2001 to 2018, using multimission remote sensing data [16]. However, the changes and driving factors of glacial lakes in this region were not analyzed from different altitudes. Currently, glacial lake research in the Altai Mountains primarily focuses on the use of high-resolution remote sensing imagery to monitor glacial lakes, and assess the potential hazards associated with them. Using high-resolution remote sensing images to monitor glacial lake changes is a fast and convenient technique, but it is limited by the cumbersome data preprocessing process, and the performance of the glacial lake information extraction algorithm. In addition, there is a paucity of studies predicting future glacial lake changes in this region.

Google Earth Engine (GEE) provides a wealth of multisource remote sensing data, which can be used to analyze the dynamic evolution of ground objects for a long time series. GEE provides cloud computing and a variety of pattern recognition algorithms to support the rapid object information extraction of remote sensing big data [18]. For this reason, we carried out a survey of the glacial lakes in the Altai Mountains using Landsat remote sensing data from 2000 to 2020, on the basis of GEE. Through the analysis of temperature and precipitation data, this paper investigated the spatial and temporal features of the glacial lakes in the Altai Mountains at various elevations, and used the MCE CA–Markov model to predict the change trend in the development of the glacial lakes in the future, so as to fill in the gaps in the existing research on the glacial lakes in the Altai Mountains in these two aspects. The research results will be helpful in developing and utilizing water resources and hazard warnings in the Altai Mountains.

## 2. Materials and Methods

### 2.1. Study Area

The Altai Mountains (Figure 1) are part of the Central Asian montane system, which runs southeast to northwest, covering the land areas of China, Mongolia, Kazakhstan, and Russia, with a total length of approximately 2000 km [19]. The glacial lakes are primarily concentrated in the northern Xinjiang region, Mongolia's western plateau, and Kazakhstan's eastern regions. The distribution of glaciers is concentrated in the area of the Katunsky, Tavan Bogd, and Chuya mountains [20]. The Altai Mountains experience a typical temperate continental climate, characterized by long, cold winters and short, hot summers. The precipitation in this region is unevenly distributed, due to the influence of the westerly circulation and the Arctic cold air mass. During the summer months, the westerly circulation transports moisture from the Atlantic Ocean through the Irtysh Valley and into the Altai Mountains, resulting in varying levels of precipitation. In winter, the westerly circulation is hindered, due to the proximity of the center of the Siberian High to the study area, and the precipitation in the western region of Mongolia decreases [21–23]. On the basis of previous studies [16,19], and according to the different regional distributions of temperature and precipitation across the Altai Mountains region, we divided the study area at 49°N into northern and southern parts, and analyzed the spatial variation features of glacial lakes in the Altai Mountains.

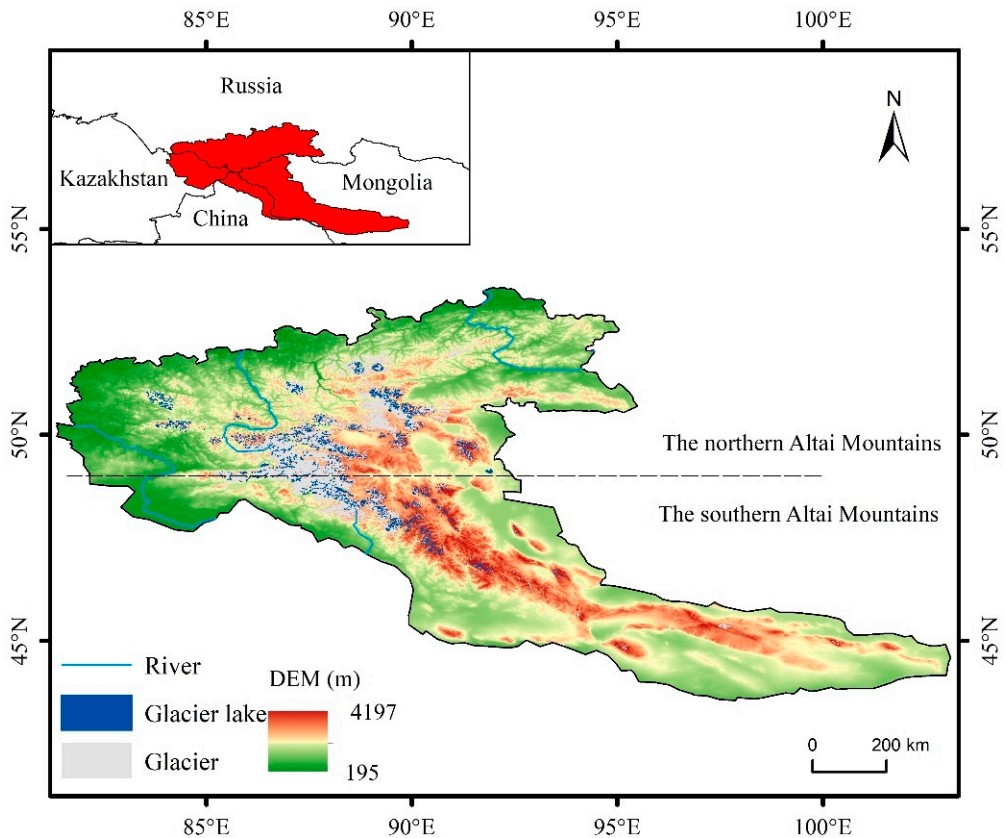

**Figure 1.** Study area and distribution of glaciers and glacial lakes in 2020.

### 2.2. Data

For this research, the GEE platform was utilized to acquire Landsat 5/7/8 TM/ETM+/OLI imagery of the Altai Mountains, spanning the period from 2000 to 2020 (Figure 2). These images were used for the extraction of information on the boundaries of glacial lakes within the study area and, subsequently, supervised classification and original image processing were performed. To reduce the snowmelt data errors arising from seasonal differences, we used images from September to October that had less cloud coverage, and that showed

stable changes [20]. Due to the excessive cloud cover of the images from 2000, 2002, 2003, and 2012, the impact on the data quality was unavoidable, so the images from August to November were used in the corresponding years.

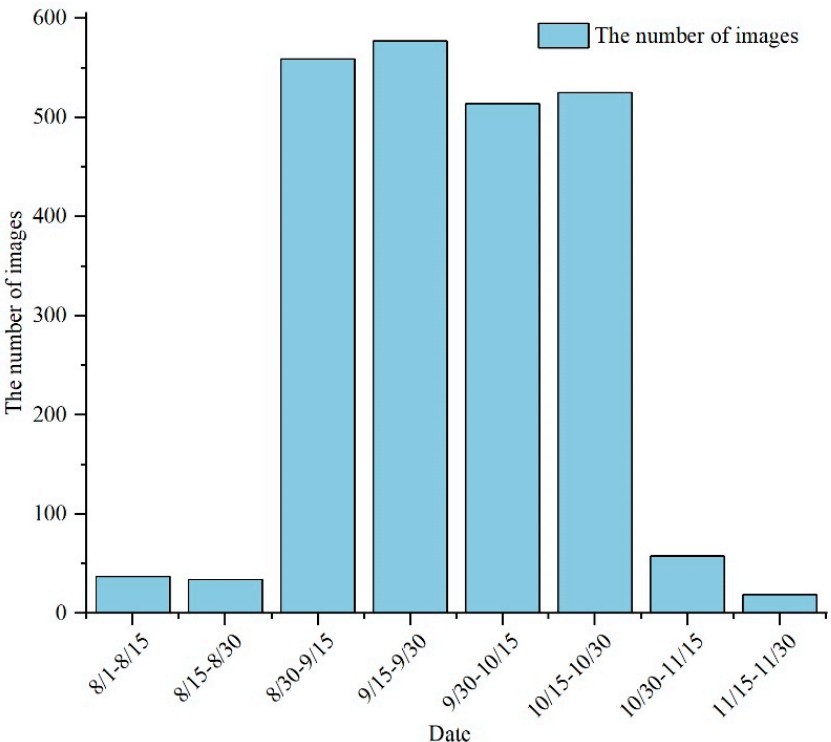

**Figure 2.** The number of images used in this study.

The digital elevation model (DEM) data were taken from the SRTM DEMUTM dataset, with a spatial resolution of 90 m. This dataset covers the extensive Altai Mountains region, including areas within China, Mongolia, Kazakhstan, and Russia. We downloaded the DEM data for 2020, and used them to extract the mountain shading, elevation, and slope for spatial analysis and for the prediction of glacial lake distribution. The data sources were collected from the Geospatial Data Cloud site, Computer Network Information Center, Chinese Academy of Sciences (http://www.gscloud.cn/), accessed in 2000.

The ERA 5 data were acquired from the fifth iteration of the European Centre for Medium Range Weather Forecasts (ECWMF) Atmospheric Reanalysis of Global Climate Data (https://cds.climate.copernicus.eu). The temperature and precipitation data have a monthly temporal resolution and a spatial resolution of 0.1°, and are utilized to analyze the underlying climate patterns associated with glacial lake changes [21]. Considering the time delay in glaciers responding to climate change [24], we selected the data from 1990 to 2020 for our analysis.

The glacier-cataloging data utilized in this study were sourced from the 2017–2018 glacier cataloging data in northwest China (http://www.csdata.org) and the Global Land Ice Measurements from Space (GLIMS) Glacier Database (https://www.glims.org/), which is last modified on 30 June 2020. The glacial lake cataloging data came from the High Asia Glacial Lake cataloging dataset (http://www.ncdc.ac.cn), accessed in 2018. We extracted the glacial lake boundaries from glacier cataloging data and Landsat TM/ETM+/OLI imagery by performing an artificial visual interpretation of the data.

*2.3. Methods*

2.3.1. Image Supervision and Classification Based on GEE

We conducted supervised classification based on the Landsat-5/7/8 remote sensing images provided by the GEE platform, using a random forest method. Seventy percent

of the sample points were used for monitoring classification, and the other 30% were used for accuracy verification. The mean OA and kappa coefficient of overall accuracy were 0.85 and 0.83, respectively, and were obtained by building a confusion matrix and performing the associated calculation.

2.3.2. Glacial Lake Boundary Extraction

We compared the supervised classification results with the original images and glacier cataloging data [22], and extracted glaciers through artificial visual interpretation. Similarly, glacial lakes were extracted through the comparing of the images with the glacial lake catalog data. Once the glacier and glacial lake boundaries were vectorized, it was necessary to remove errors caused by the existence of mountain shading. This step relied primarily on comparisons with the existing cataloging data, and manual visual interpretation. Based on existing methods, we set a buffer length of 10 km centered on the glacier terminus [23], and removed lakes with an area of less than 0.01 square kilometers [25], to perform the secondary refinement of the frozen lake surface. The overall error in estimating the area and number of glacial lakes was 9.14%, when compared to the data from the Asian high glacial lake catalog, which was within acceptable limits.

2.3.3. Sen + Mann–Kendall Trend Analysis

Theil–Sen slope estimation (henceforth referred to as Sen) and the Mann–Kendall significance test are two nonparametric testing procedures [26], and estimation of the Sen slope is employed in the process of determining the value of the trend. As a result of the suggested method's great computing efficiency, as well as its insensitivity to measuring error and outliers, it is frequently utilized in the process of trend analysis of large time-series data. The formula for the computation is given as follows:

$$\beta = \text{Median}\left(\frac{x_j - x_i}{j - i}\right), \forall j > i \tag{1}$$

The degrees $\beta$ of positive and negative trends can represent the rise and fall of the time series. When analyzing a time series, the value of $\beta$ larger than 0 shows that the series has an upward trend, and vice versa.

It is common practice to integrate the Mann–Kendall nonparametric test with the Sen slope estimate when computing the trend value. In other words, the Sen trend is computed at the beginning, and then the Mann–Kendall technique is applied, to determine whether the trend is significant. The calculation method of the Mann–Kendall test statistic S is as follows:

$$S = \sum_{i=1}^{n-1} \sum_{j=i+1}^{n} \text{sgn}(x_j - x_i) \tag{2}$$

$$\text{sign}(x_j - x_i) = \begin{cases} +1, & x_j - x_i > 0 \\ 0, & x_j - x_i = 0 \\ -1, & x_j - x_i < 0 \end{cases} \tag{3}$$

where n denotes the number of data in the sequence, $x_i$ and $x_j$ signify the data values in time series i and j (j > i), and the sign ($x_j - x_i$) indicates the sign function [27].

The temperature and precipitation data spanning the period from 1990 to 2020 were chosen for the analysis of the driving factors, using n = 31. As all stations have over 10 years of recorded data, Formula (4) gives the following relationship, which is utilized to calculate the variance as follows [28]:

$$\text{VAR}(S) = \frac{(n(n-1)(2n+5) - \sum_{i=1}^{m} t_i(t_i - 1)(2t_i + 5))}{18} \tag{4}$$

where m denotes the number of nodes in the sequence, and $t_i$ represents the width of the node.

Then, we used the test statistic Z to conduct the test for trend, and took the level of significance as $\alpha = 0.05$, $Z_{(1-\alpha)} = Z_{0.975} = 1.96$. The formula for computing the Z value is:

$$Z = \begin{cases} \frac{S-1}{\sqrt{\text{VAR(S)}}}, S > 0 \\ 0, S = 0 \\ \frac{S-1}{\sqrt{\text{VAR(S)}}}, S < 0 \end{cases} \tag{5}$$

### 2.3.4. MCE CA–Markov

We coupled CA with Markov models, and utilized the matrix of transition probabilities to model an alteration in the pattern of land cover over time, by employing the CA–Markov model, which is part of the IDRISI software package [29]. Firstly, we acquired the transition probability matrix of ground-cover type changes in the research region from 2000 to 2010, 2010 to 2015, and 2010 to 2020, by making use of the Markov module, which is included in the IDRISI program.

$$P_{ij} = \begin{bmatrix} P_{11} & \cdots & P_{1n} \\ \vdots & \ddots & \vdots \\ P_{n1} & \cdots & P_{nn} \end{bmatrix} \tag{6}$$

The proportional error parameter of the model influences the precision of the simulation by affecting the transition probability matrix. When the proportional error parameter is 0.15, the classification accuracy of images can generally reach over 85% [30]. Therefore, we set the proportional error parameter to 0.15 in the Markov process of this model.

Land-cover-type transfer suitability atlases refer to the probability map of a land-cover type transitioning to other land-cover types, and are an important component of conversion rules [31]. To explore the changes and driving factors of glacial lakes in the Altai Mountains at different altitudes, we selected four driving factors that affect land-cover change in the study area: the elevation, slope, temperature, and precipitation. The slope was obtained from the DEM data of the study area in 2020 by surface analysis in ArcGIS 10.8. The average rainfall and temperature data between 1990 and 2020 have been computed for the production of the suitability atlas. As we selected DEM data with a spatial resolution of 90 m, the land-cover classification data and the above driving factor data should be resampled to the same spatial resolution (90 m), to ensure the normal operation of the prediction model. Through the application of the Boolean multiplication method within the multicriteria assessment (MCE) module of the IDRISI software, we merged constraints from a variety of object types, and suitable images of various land-cover types were obtained. Finally, we used the Collection Editor module within the software package to synthesize the atlas of the transformation of land-cover-type suitability across the study region.

Once the *p* value had been determined, the following expression could be used to predict the future land-cover type:

$$S_{(t+1)} = P_{ij} \times S_{(t)} \tag{7}$$

where $S_{(t)}$ and $S_{(t+1)}$ are representations of the land-cover states at times "t" and "t + 1", respectively, and $P_{ij}$ denotes the transition probability matrix of the land-cover type.

We imported the Markov transition matrix, the map of land-cover types in the Altai Mountains area, and the adequacy atlas produced under the IDRISI software's CA–Markov module. Referring to existing research [32,33], we set the cycle count to ten, and then used the default Moore $5 \times 5$ neighborhood filter to obtain the simulated image.

In contrast to the approach to previous tests with a single Kappa coefficient [34–36], as shown in Table 1, we extended the Kappa coefficient validation to the standard Kappa coefficient (Kstandard), random Kappa coefficient (Kno), location Kappa coefficient (Klocation), and stratified location Kappa coefficient (KlocationStrata) in the VALIDATE module of

IDRISI Selva17.00, to more comprehensively test the degree of consistency between the simulation and the actual image, in terms of the quantity and position [37,38].

**Table 1.** Kappa coefficient test series group.

| Name | Formula | Meaning |
| --- | --- | --- |
| Kstandard | $Kstandard = \frac{P_0 - N(m)}{1 - N(m)}$ | The expectation value of N (m), which has the ability to maintain a certain number, but not the ability to maintain spatial location, is used to evaluate the comprehensive information change. |
| Kno | $Kno = \frac{P_0 - N(n)}{1 - N(n)}$ | The expected value of N (n), which has neither the ability to maintain quantity, nor the ability to maintain spatial location, is used to evaluate the comprehensive information change. |
| Klocation | $Klocation = \frac{P_0 - N(n)}{P(m) - N(n)}$ | It is assumed that N (n), capable of keeping some quantity, but unable to keep space position, can be used as an expectation value, and that P (m), which can keep some quantity and space position completely, is considered as a true value for the evaluation of space-position information. |
| KlocationStrata | $KlocationStrata = \frac{P_0 - N(n)}{K(m) - N(n)}$ | N (n), which has neither the ability to maintain quantity, nor the ability to maintain spatial location, is taken as the expected value, and K (m), which has both the ability to maintain quantity and the ability to completely maintain the spatial layer/region, is taken as the true value to evaluate the change in spatial location information. |

### 2.3.5. Pearson Correlation Coefficient Method

The technique for calculating the Pearson correlation coefficient was used to conduct the research necessary to determine the extent of the connection that exists among glaciers, precipitation, temperature, and glacial lake alteration. Typically, the symbol r is used to signify the sample correlation coefficient, while the symbol p is employed to indicate the population correlation coefficient. The overall correlation coefficient formula is:

$$\rho_{XY} = \frac{Cov(X, Y)}{\sqrt{VAR(X)}\sqrt{VAR(Y)}} \tag{8}$$

where Cov(X,Y) signifies the covariance of the random variables X and Y, and VAR(X) and VAR(Y) are representations of the variances in X and Y, respectively.

Letting $X_1, X_2 \ldots X_n, Y_1, Y_2 \ldots Y_n$ be a set of simple random samples from the X and Y populations, respectively, the sample correlation coefficient can be calculated using the following formula:

$$r = \frac{\sum_{i=1}^{n}(X_i - \overline{X})(Y_i - \overline{Y})}{\sqrt{\sum_{i=1}^{n}(X_i - \overline{X})^2 \sum_{i=1}^{n}(Y_i - \overline{Y})^2}} \tag{9}$$

Using the observed value, we can compute the sample correlation coefficient, and a greater absolute value of r indicates a stronger relationship between the two variables [39].

## 3. Results

### 3.1. Glacial Lake Distribution and Changes

Statistical analysis was conducted to investigate the changes in the area and number of glacial lakes in the Altai Mountains between 2000 and 2020 (Figure 3). The results indicated an increase in both the area and number of glacial lakes during this period. The total area of the glacial lakes was 582.90 km$^2$ in 2000, and increased to 682.38 km$^2$ in 2020, representing a growth rate of 17.07%. For 21 years, the number of glacial lakes increased from 2587 to 3824, with a growth rate of 47.82%. The speed at which these glacial lakes grew was faster than that of their total area over the span of 21 years.

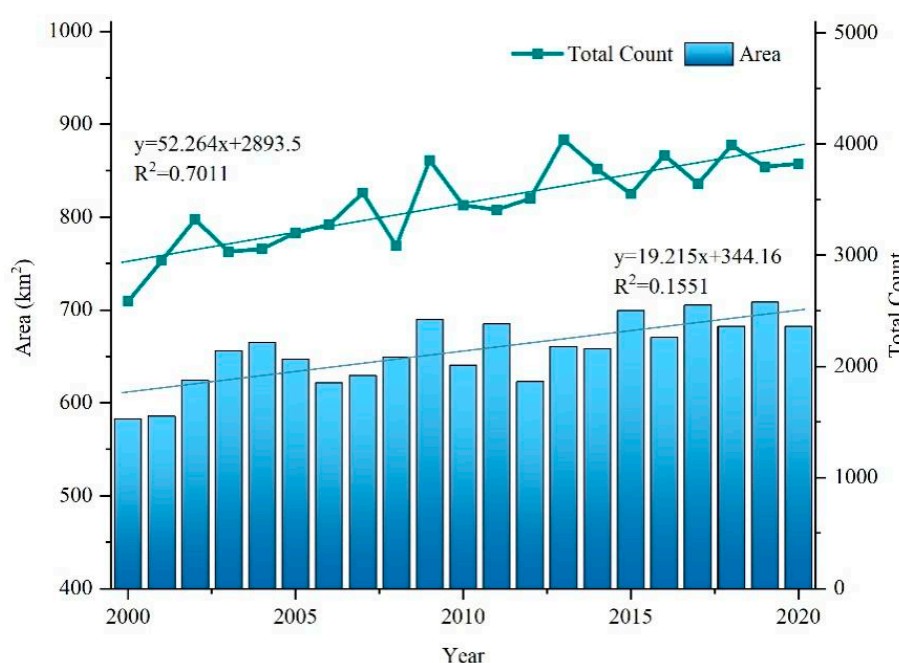

**Figure 3.** The total number of glacial lakes, and the area within the study area, from 2000 to 2020.

The overall view of glacial lake change is that both area and number show an increasing trend. However, the changes did not increase in stages, but fluctuated irregularly. Of the 21 years of data, the period between 2007 and 2015 was the period with the most obvious fluctuation. Considering 2007–2010 as a typical year, the quantity of glacial lakes decreased by 474 between 2007 and 2008, and then rapidly increased to 3857 between 2008 and 2009, which became the period with the largest increase in the number of glacial lakes in the 21 years. Taking the period from 2010 to 2012 as an example, the area of the glacial lake witnessed a significant change. Specifically, it increased by 44.10 $km^2$ from 2010 to 2011, and then decreased to 622.95 $km^2$ from 2011 to 2012. This period represents the most substantial fluctuation in glacial lake area throughout the 21-year data. Since 2014, there has been a slowdown in the fluctuations in both the area and number of glacial lakes. The average surface area of glacial lakes during the seven-year period is approximately 686.86 $km^2$, while the number of glacial lakes has remained relatively stable, at approximately 3783.

Figure 4 and Table 2 reflect the distribution of glacial lakes in the northern and southern Altai Mountains. There are more glacial lakes in the north than in the south. There were 1571 glacial lakes in the north in 2000, accounting for 60.73% of the total. The total amount of glacial lakes in the north grew at an average pace of 51 each year. Up to 2020, the number of glacial lakes has risen to 2591, which constitutes 67.76% of the total. The northern part of the Altai Mountains has many modern glaciers, and the relief is relatively low at the glacier margins [40–42], which is a favorable environment for glacial lake growth and development. Regarding the proportion of glacial lake area, the original amount in the south was 30.06% in 2000, and rose to 45.11% in 2020. In contrast, the northern region's proportion of glacial lake surface decreased from 69.94% in 2000 to 54.89% in 2020. A closer comparison shows that, while many new glacial lakes have been produced in the northern Altai Mountains over the last 21 years, their area decreased by approximately 8.12%, compared to the area in 2000. The number of glacial lakes in the southern Altai Mountains improved at a lower rate than that in the north (10.85 per year on average), but the surface area of glacial lakes in the south increased by 75.67%, relative to that of 2000. In general, although there are few new glacial lakes in the southern Altai Mountains, the area of glacial lakes is growing rapidly, and there is potential for extensive expansion in the future.

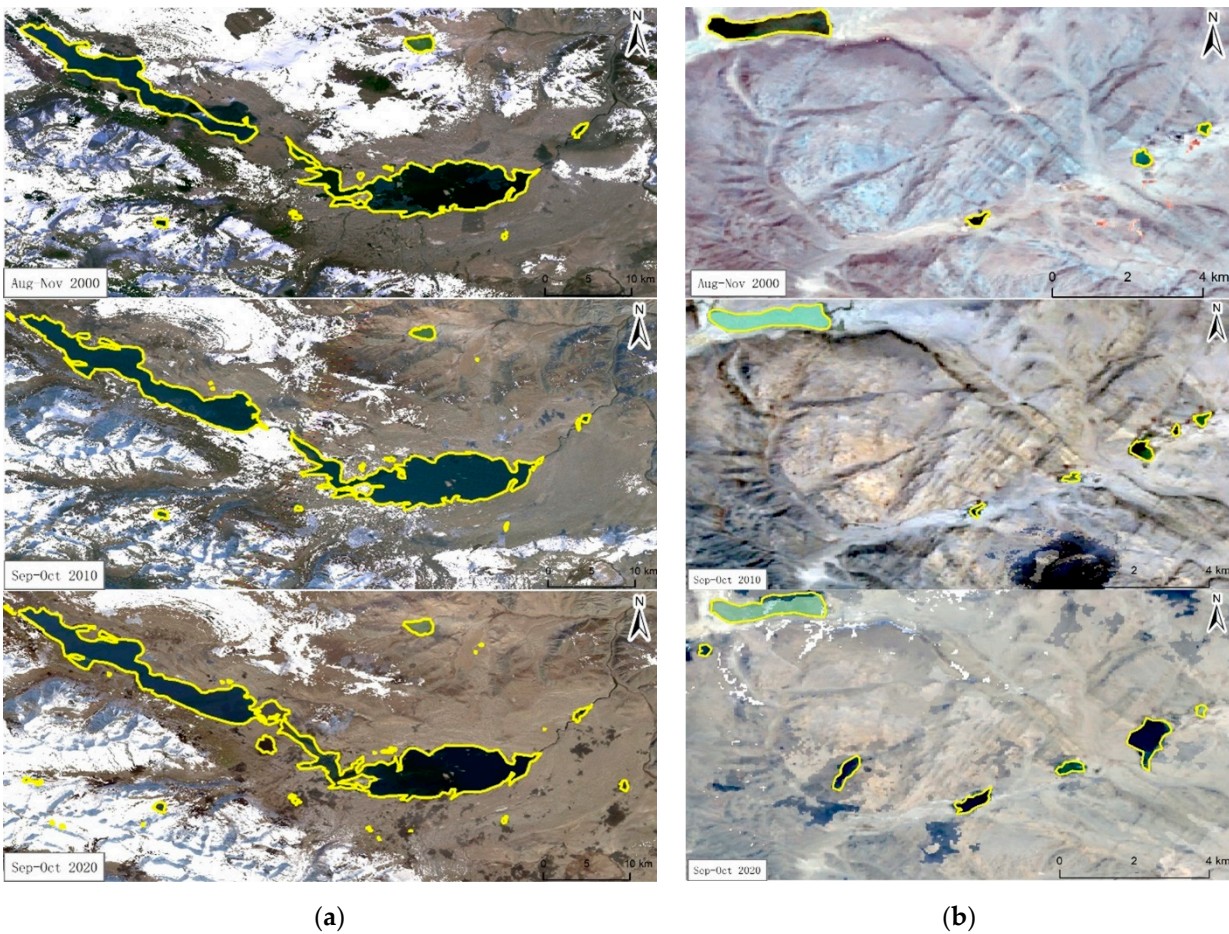

(**a**)    (**b**)

**Figure 4.** Glacial lake inventories from various parts of the study area in 2000, 2010, and 2020: (**a**) a typical area in the northern Altai Mountains; (**b**) a typical area in the southern Altai Mountains.

**Table 2.** Glacial lake inventories from various parts of the study area at five timepoints from 2000 to 2020.

| Year | The Northern Part | | The Southern Part | |
|---|---|---|---|---|
| | Count | Area (km$^2$) | Count | Area (km$^2$) |
| 2000 | 1571 | 407.67 | 1016 | 175.23 |
| 2005 | 1971 | 430.25 | 1231 | 217.01 |
| 2010 | 2372 | 368.52 | 1078 | 272.32 |
| 2015 | 2407 | 434.43 | 1145 | 265.32 |
| 2020 | 2591 | 374.55 | 1233 | 307.83 |

*3.2. Characteristics of Various Sizes of Glacier Lakes*

Glacial lake areas in the Altai Mountains have varied between 0.01 and 9.85 km$^2$ over the last 21 years. Based on a previous study [43,44], considering the distribution of glacial lakes in various regions of the study area, we categorized the glacial lakes into different groups: <0.1 km$^2$, 0.1–0.2 km$^2$, and >0.2 km$^2$. This classification facilitates investigation into the different scale changes in the Altai Mountains glacial lakes (Figure 5). In this region, the majority of glacial lakes have a surface area below 0.1 km$^2$, comprising 41.13% of the total. Conversely, larger glacial lakes, exceeding 0.2 km$^2$, constitute a significant portion of the overall glacial lake area, accounting for 70.21% of the total area.

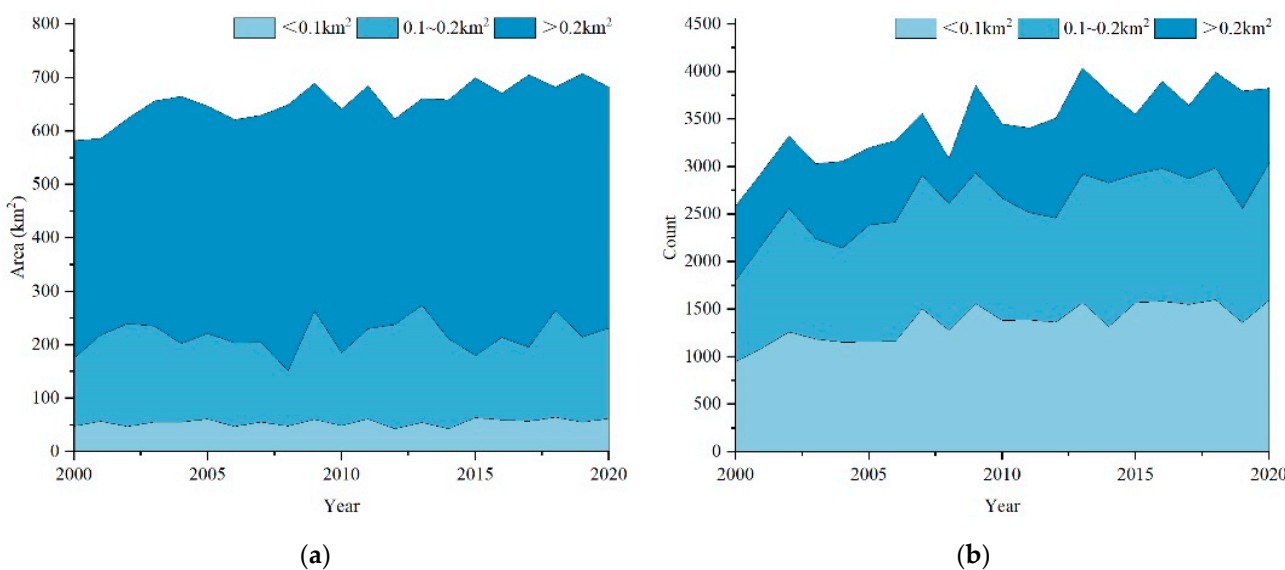

**Figure 5.** Area and quantity of glacier lakes in different size classes: (**a**) glacial lake areas during the 21 years studied; (**b**) glacial lake numbers in the 21 years studied.

From 2000 to 2020, a total of 650 new glacial lakes, each with an area below 0.1 km$^2$, were formed, constituting 52.55% of the overall new glacial lakes. Notably, there was a marked rise in the size of glacial lakes ranging from 0.1 km$^2$ to 0.2 km$^2$, at a rate of 2.09 km$^2$ annually on average, accounting for 42.04% of the overall growth in the glacial lake area. Glacial lakes greater than 0.2 km$^2$ in area have decreased in quantity from 788 lakes in 2000 to 786 lakes in 2020. It can be concluded that small-scale glacial lakes in the Altai Mountains are sensitive to changes in the climate and the surrounding environment, and the changes are more significant, which is in line with Chen and his colleagues' opinion that small-scale glacial lakes in High Asia show the most obvious growth [45]. Large-scale glacial lakes have a stronger storage capacity, and are not as likely to change in response to external factors.

### 3.3. Characteristics of the Elevational Distribution of Glacial Lakes

In the study area, glacial lakes are found within an elevation range of 1400 m to 4000 m, exhibiting a varied distribution across this elevation gradient, and we split this elevation range into 100 m intervals (Figure 6). From the results, glacial lakes are predominantly distributed between 2000 m and 3000 m above sea level, constituting 59.96% of the total. The elevation distribution range of glacial lakes in the Altai Mountains is lower than that of glacial lakes on the Tibetan Plateau (4400–5400 m) [46] and in the Tianshan Mountains (2900–4100 m) [47]. Therefore, we can assume that the higher the latitude, the lower the altitude of the glacial lake distribution. Higher elevations exhibit a decreasing number of glacial lakes. As high as 3600 m above sea level, there are only 26 glacial lakes in this region, and among the total number of glacial lakes, a mere 0.68% represent this specific category, while their average surface area constitutes only 0.24% of the overall glacier lake surface area. The upward trend in glacial lake expansion primarily occurs within the 2000 m to 2500 m altitude range. In 2020, this region contained 1143 glacial lakes, marking an increase of 569 compared to the 2000 count. In 2020, the glacial lake area within the specified elevation range of the study area measured 199.81 km$^2$, an increase of 128.98 km$^2$ from the 2000 level. Furthermore, our findings, along with those in Figure 6 and an assessment of climate change in the study area, indicate that, as the temperature reached its lowest point in 2010, the Altai Mountains region underwent a decline in its snowline elevation. Consequently, the glacial lakes in the region displayed a propensity to expand at lower elevations.

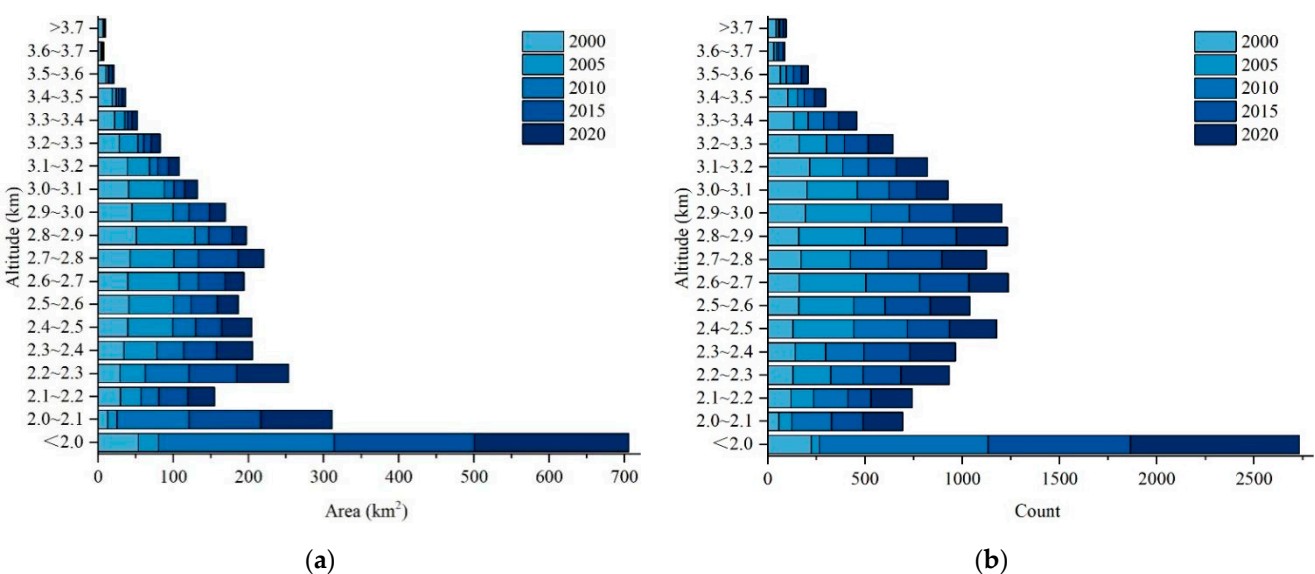

**Figure 6.** The altitudinal distribution of glacial lakes, and changes from 2000 to 2020: (**a**) area distributions and changes in glacial lakes; (**b**) quantity distributions and changes in glacial lakes.

### 3.4. Analysis of the Driving Force Affecting Glacial Lake Change

The growth and disappearance of glacial lakes are strongly influenced by climate change, particularly temperature and precipitation. These factors play a role in the glacier expansion and retreat, surface runoff, and evaporation rates. Additionally, they impact the water supply to the glacial lake, depending on the water volume within the lake basin, and the basin's structure [48–54]. Considering the time delay for glaciers to respond to climate change [24], our analysis focused on temperature and precipitation variation in the Altai Mountains from 1990 to 2020.

#### 3.4.1. The Overall Pattern of Climate Change

Figure 7 depicts the general climate trends in the Altai Mountains from 1990 to 2020. Overall, the temperature in the region shows stable fluctuations, and the period of fluctuation gradually shortens. There were significant increases in temperature in 1997, 2002, and 2007, and significant decreases in 1993, 1996, and 2010. Prior to 2013, the abrupt rise and fall in temperature alternated, and the fluctuation was mild after 2014, maintaining the overall temperature curve of the Altai Mountains area. The overall rainfall in the study area exhibits an increasing trend in annual rainfall changes. Except for during a few years, such as 1997, 2008, and 2011, the majority of rainfall in the area has surpassed, or has been closely aligned with, the average rainfall over the 31-year period.

#### 3.4.2. Sen + Mann–Kendall Test of Temperature and Precipitation Trends

The Altai Mountains, situated at the middle and high latitudes, experience a typical temperate continental climate for the Eurasian continent. Cold, moist air originating from the Arctic Ocean penetrates the northwestern region of the Altai Mountains through the upper Ertysh Valley, and is uplifted by the terrain, resulting in precipitation. In the southern and southeastern parts of the research area, due to the deep inland topography, it is difficult for water vapor to reach this area due to the blocking terrain, resulting in little precipitation in this area [19,55–57].

A map of the precipitation change trends in the Altai Mountains region was created using the Sen + Mann–Kendall method (Figure 8). In the northern Altai Mountains, the terrain is relatively flat, which is suitable for glacial lake growth and development. Since 2011, there has been a consistent rise in rainfall levels observed in the northern part of the study area. In 2020, 298 new glacial lakes were established in the region, and most of them had an area of less than 0.2 km$^2$. In the southern Altai Mountains, precipitation is

concentrated mainly in the central and southwestern areas, while higher elevations in the southeast experience reduced precipitation.

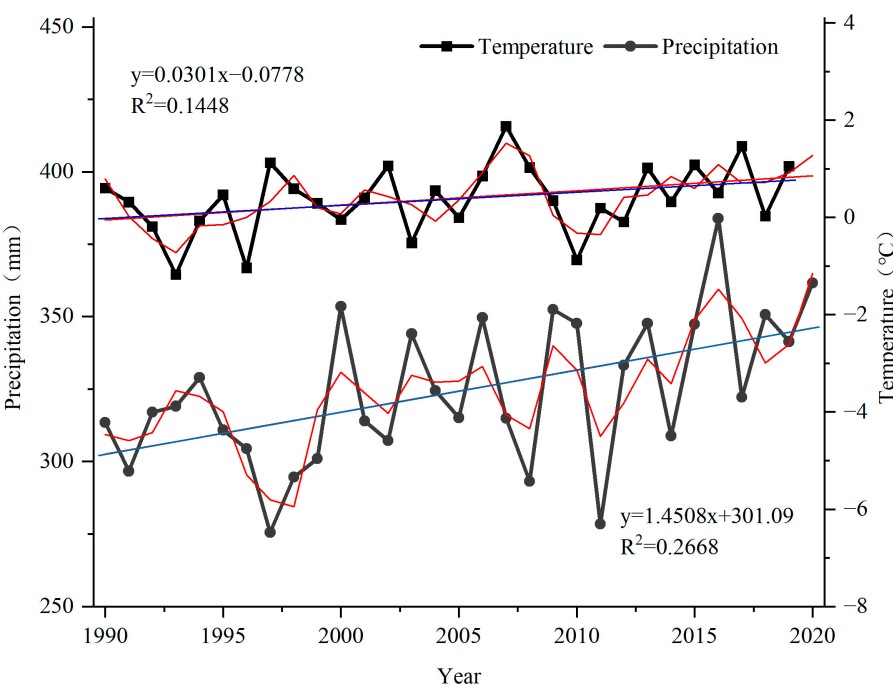

**Figure 7.** Changes in temperature and precipitation in the Altai Mountains from 1990 to 2020. The linear regression is represented by the blue line. The moving mean is represented by the red curve in the graph.

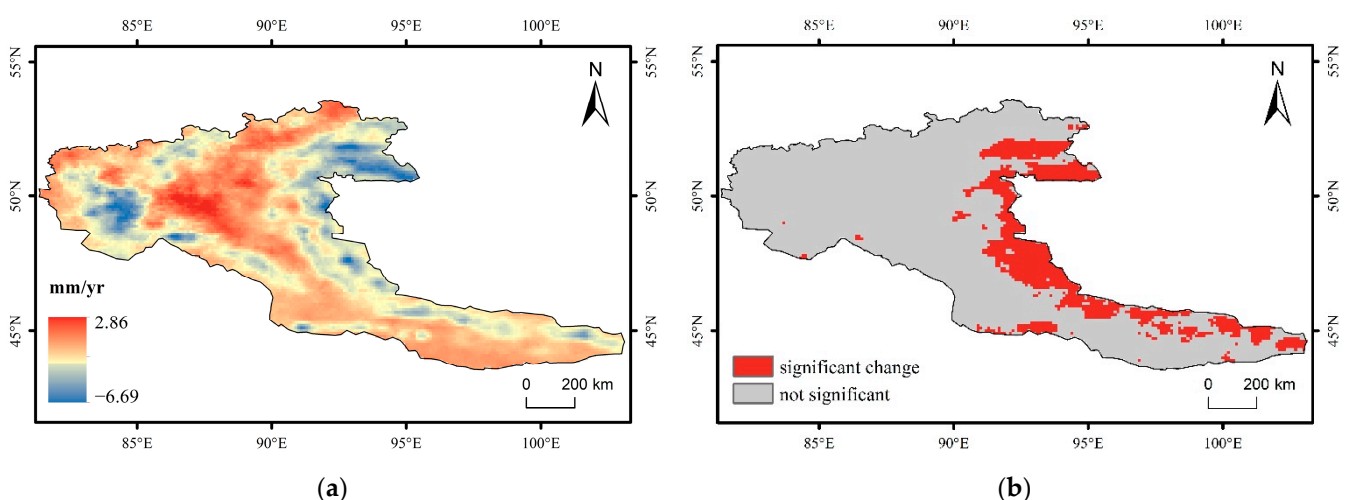

(**a**)                    (**b**)

**Figure 8.** (**a**) The spatial trends in precipitation across the study area; (**b**) the results of the Mann–Kendall test.

The average global temperature has generally increased in recent years. In this context, a large area of the glacier is in the process of melting, and the glacial lake at the glacier terminus is directly affected [11,58]. This map of temperature change trends may reflect the temporal and spatial features of the temperature evolution in the Altai Mountains region (Figure 9). Figure 9 shows that the temperature in the southern Altai Mountains region is generally increasing, and the temperature in a wide area of the northern region is slightly reduced. In the north, there are more lower-altitude areas than in the south, and these regions are concentrated near glacial areas, which satisfy the conditions for glacial lake development; thus, the number of glacial lakes in the northern region surpasses the count

in the southern region. The rainfall in the research area decreased from 2006 to 2007, but as temperatures in this region have continued to rise since 2005, continued warming has led to the melting of many glaciers in the north, meeting the needs of expanding glacial lakes. Thus, as the southern glacial lakes have experienced a notable decline in numbers, the northern glacial lakes have witnessed an increase, reaching a total of 858. Although the south has a relatively low number of glacial lakes (117), their area has expanded, due to a slight increase in rainfall and a substantial increase in temperature. By 2020, the surface area of glacial lakes had expanded by approximately 28.70 km$^2$.

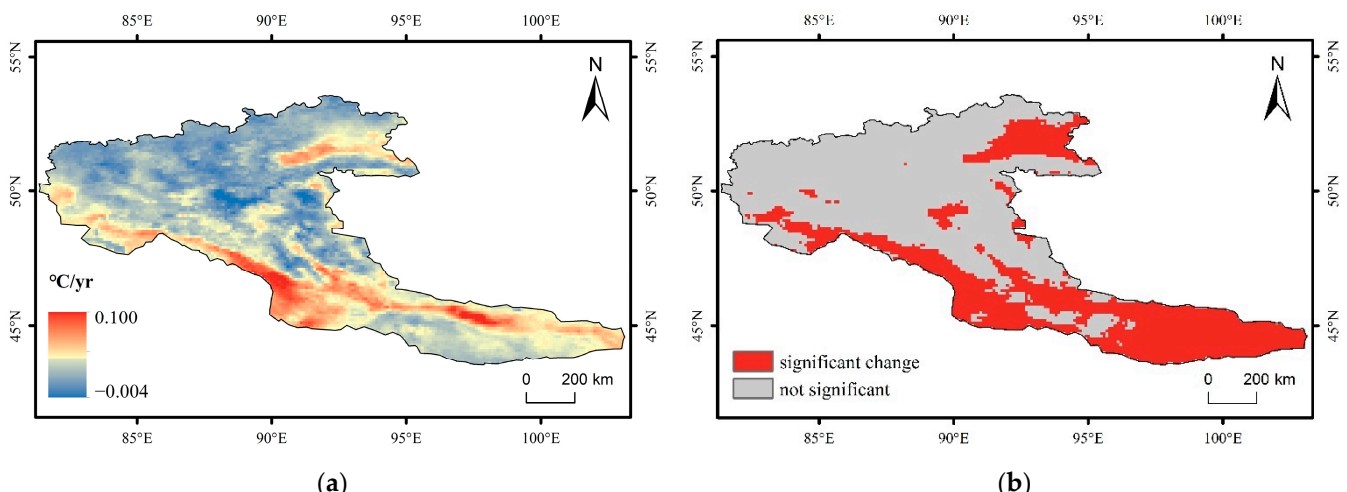

(**a**)　　　　　　　　　　　　　　　　　　(**b**)

**Figure 9.** (**a**) Spatial temperature trends across the study region; (**b**) the results of the Mann–Kendall test.

### 3.5. Land-Cover Type Changes

Following the classification and summary of the land-cover transfer matrix from 2000 to 2010, 2010 to 2015, and 2015 to 2020 in the Altai Mountains area; from 2000 to 2020, we acquired both the area and the evolution of the various land-cover categories within the study area (Figure 10).

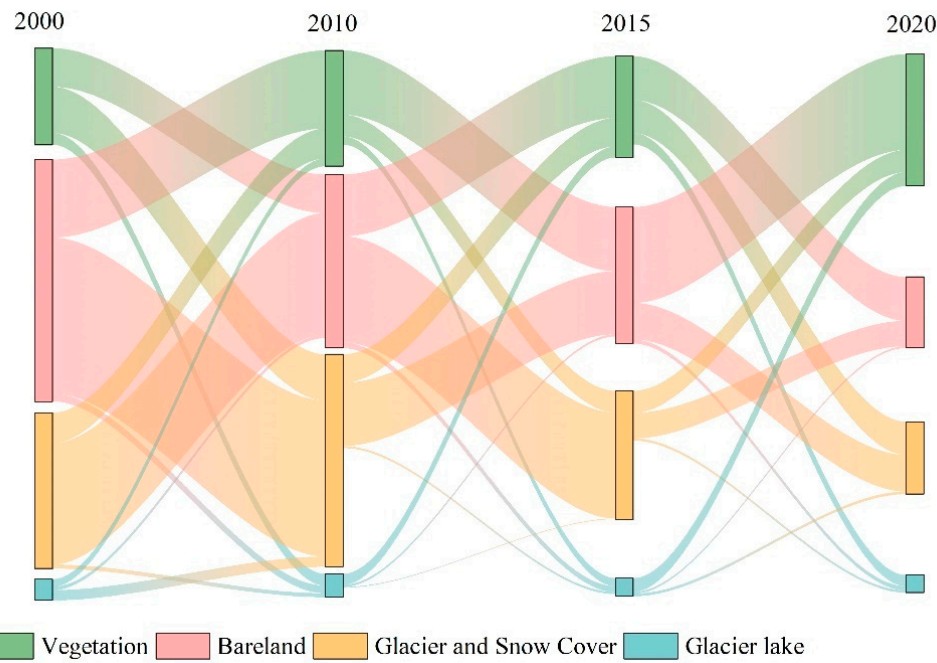

**Figure 10.** Map of Sankey land-cover transfers in 2000–2010, 2010–2015, and 2015–2020.

Vegetation, bare land, glaciers, snow cover, and glacial lakes are the main types of land cover in the Altai Mountains. The analysis of area changes reveals that the Altai Mountains experienced significant increases in glacier and snow cover, compared to other land-cover types, between 2000 and 2015, increasing by 1507.68 km$^2$ and 923.15 km$^2$ from 2000 to 2010, and 2010 to 2015, respectively. Vegetation was the land-cover type with the greatest increase in area between 2015 and 2020, with a total of 3498.08 km$^2$, and most of its expansion came from bare land. From 2000 to 2015, a continuous growth trend was calculated in the area of the glacial lakes, with increases of 10.52 km$^2$ and 6.04 km$^2$, respectively. Vegetation and bare land contributed 50.29~51.22% and 32.72~35.85% to the increase in glacial lake area, respectively. Between 2015 and 2020, the glacial lakes saw a 17.36 km$^2$ drop in surface area. During this period, the region that had previously been covered by glacial lakes began to be covered by vegetation. Since 2014, the fluctuations in temperature and precipitation have weakened. The vegetation in the study area exhibited an increasing trend under favorable hydrothermal conditions.

### 3.6. Future Glacier Lake Trends

The kappa coefficients for the multicriteria CA–Markov model are greater than 0.75 (82.88%, 85.45%) when the time interval is 5 years and 10 years. The test values of the expanded four kappa coefficients are as follows: the Kstandard coefficients were 0.8064 and 0.8300, the Kno coefficients were 0.8052 and 0.8884, the Klocation coefficients were 0.8267 and 0.8539, and the KlocationStrata coefficients were 0.8267 and 0.8539, respectively. These findings indicate that the model simulation provides accurate results, and the fidelity is high. This simulation is appropriate for modeling the change in land-cover types across the study region in 2025 and 2030 (Figure 11).

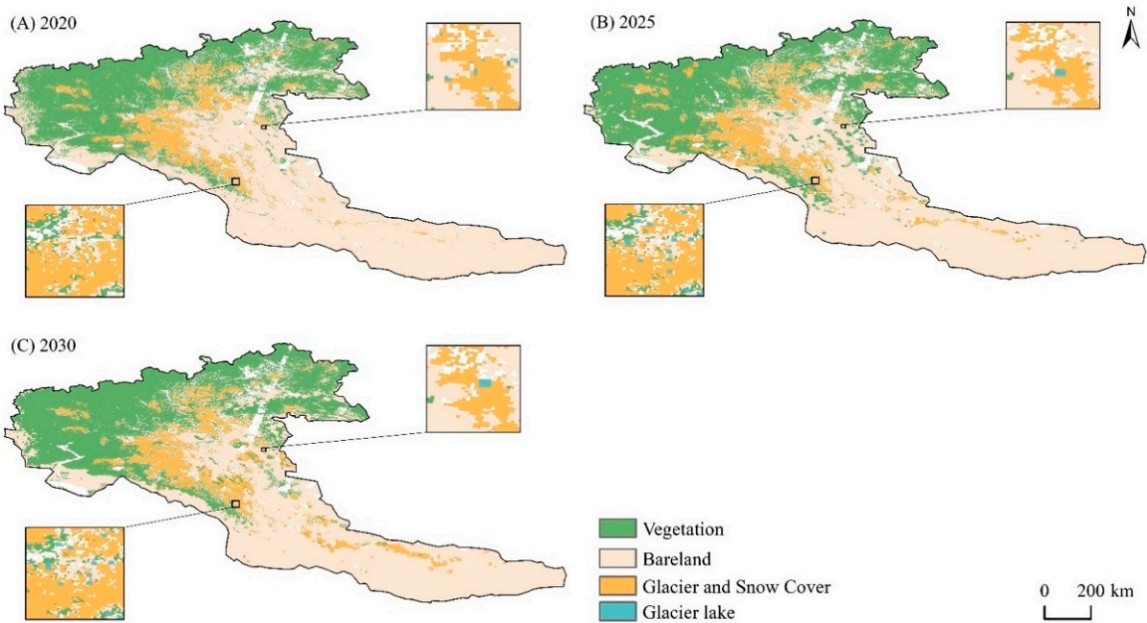

**Figure 11.** Prediction of the spatial distribution of land cover in the Altai Mountains in 2020, 2025, and 2030.

According to statistics, the area and quantity of glacial lakes are expected to rise to 752.70 km$^2$ and 3946 by 2025, increasing by approximately 10.30% and 3.19%, respectively, compared to 2020. The area and quantity of glacial lakes are projected to increase to 793.78 km$^2$ and 4111 by 2030, increasing by 14.59% and 16.55% from 2020, and increasing by 5.48% and 4.18% from 2025, respectively.

Based on further analysis (Table 3), by 2025, the projected area of glacial lakes in the northern Altai Mountains is 408.73 km$^2$, while in the southern Altai Mountains, it is estimated to be 343.96 km$^2$. The area of glacial lakes in the northern region is projected

to increase by approximately 9.13% by 2025 compared to 2020, while the area of glacial lakes in the southern region is expected to grow by 11.74%. Despite the northern region having a larger glacial lake area, the southern region demonstrates faster growth rates from 2020 to 2025. By 2030, the northern Altai Mountains glacial lake area will increase to 428.73 km$^2$, and the number of glacial lakes will increase to 2730. In comparison, the southern Altai Mountains will have an area of approximately 365.24 km$^2$, and a total of 1381 glacial lakes. Compared to the 2025 baseline, the northern region of the study area underwent a 4.89% increase in glacial lake area, while the southern region experienced a 6.19% increase. Additionally, the number of glacial lakes increased by 3.64% in the north, and 5.26% in the south. In general, the area of glacial lakes in the southern Altai Mountains will continue to increase rapidly over the next decade. This point warrants our attention, so that policy-making departments can carry out the development and utilization of glacial lake water resources, and provide disaster warnings in the southern Altai Mountains in the future.

**Table 3.** Glacial lake inventories in various parts of the study area in 2020, 2025, and 2030.

| Year | The Northern Part | | The Southern Part | |
|---|---|---|---|---|
| | Count | Area (km$^2$) | Count | Area (km$^2$) |
| 2020 | 2591 | 374.55 | 1233 | 307.83 |
| 2025 | 2634 | 408.73 | 1312 | 343.96 |
| 2030 | 2730 | 428.73 | 1381 | 365.24 |

Based on the results of the forecast, we believe that, after 2014, the hydrothermal conditions will gradually stabilize, and glacial lakes in the Altai Mountains will continue to occupy vegetated areas, as well as a portion of the land that is bare. However, because of the small interannual temperature variation, the level of melting of glaciers has decreased, the phenomenon of a sudden decrease in rainfall followed by a large increase in rainfall has rarely occurred since 2017, and the expansion and development of glacial lakes have been affected by the slowdown in the growth of their major water sources. Consequently, while the area and number of glacial lakes will continue to increase in the coming decade, the pace of development is gradually slowing. Luo et al. suggested that most of the glacial lakes in the Altai Mountains showed a stable or slight change trend [16]. Our study indicates that this trend will be maintained in the future for the glacial lakes in the region, under the current climate background.

During the process of using the forecast model to simulate the land-cover situation in the Altai Mountains area, in this paper, parameters were only fixed based on experience and historical land-cover change characteristics, without the consideration of the influence of climatic factors and human activities on land-cover types under different future scenarios. In addition, the difference between the simulation results at different resolutions is also worth discussing. Thus, the means of building a more complete and scientific prediction model remains a topic for further exploration.

## 4. Discussion

### 4.1. The Effect of Glacier Changes on Glacial Lakes

Glacier retreat provides ample water supply and space for subsequent glacial lake expansion, leading to rapid increases in the number and surface area of glacial lakes [59,60]. From 2000 to 2020, our analysis of data from the glacier catalog and manual visual interpretation revealed an average annual increase in the glacier area in the Altai Mountains region of 21.01%, and glacier numbers declined at an average rate of 1405.15 annually. In the Altai Mountains, glacial lakes exhibited an average annual area increase of 4.97%, and a corresponding average annual growth of 61.85 in number. However, the rate of change in the glacier area and number of glaciers surpassed that of the glacial lake area and number.

This study aimed to analyze the correlation between glaciers, precipitation, and temperature, and its impact on changes in glacial lakes, using the Pearson correlation

coefficient method. The correlation analysis (Table 4) revealed a negative correlation between the glacier area and glacial lake area, as well as between the number of glaciers and the number of glacial lakes. Despite the small absolute values of the correlation coefficients between glacial lakes and climatic factors, of the several factors chosen in this study, the correlation coefficients between the number of glaciers and the area and number of glacial lakes were 0.867 and 0.676, respectively, which represents the most significant factor for glacial lakes. In short, the influence of temperature on glacial lake change is indirect, primarily affecting the change in the glacier surface area; that is, an increasing temperature will cause glaciers to melt, subsequently leading to changes in glacial lakes [61,62]. In terms of the correlation between rainfall and the number of glacial lakes, a correlation coefficient of 0.378 indicates a positive but moderate relationship, which is more significant than the coefficient for temperature. The main reason for the increase in glacial lakes in the central Himalayas is rising temperatures [63]. The glacial lake expansion rate in the western Nyainqêntanglha Range is affected by both temperature and precipitation [64]. The expansion of glacial lakes on the Qinghai–Tibet Plateau is mainly driven by the increase in glacial meltwater, which is restricted by precipitation and evapotranspiration [65]. Rising temperatures in the Tianshan Mountains have led to the rapid retreat of glaciers, and increasing glacial lakes [15]. This phenomenon is the one most similar to the driving mechanism of glacial lake change in the Altai Mountains identified in this study. It can be concluded that glacial meltwater is the main driver of glacial lake expansion with increasing latitude.

**Table 4.** The correlation between glacial lakes and glaciers and several climate factors.

| Glacial Lake | Glacier | | Climate Factors | |
|---|---|---|---|---|
| | **Count** | **Area** | **Precipitation** | **Temperature** |
| Count | −0.676 ** | 0.065 | 0.378 | 0.287 |
| Area | −0.867 ** | −0.136 | 0.171 | 0.25 |

** $p < 0.01$.

Prior studies have shown that with increasing temperatures, increased meltwater from glaciers in the Altai area will counteract the negative effects of elevated evapotranspiration, leading to some lakes expanding [16], which is consistent with our conclusion. However, the glacial lake area decreased by 29.10 km$^2$ during 2015–2016, while the glacial surface area maintained a decreasing trend. Due to the intricate climate patterns and the non-enclosed water circulation system in the study region, it is necessary to conduct further investigation into the impact of glacial meltwater on glacial lakes.

### 4.2. Response of Glacial Lakes to Climate Change

Given that there is no significant correlation between global glacial lakes and climatic factors in the Altai area, considering the spatial heterogeneity of glacial lake change, we divided the elevational range of the study area into 100 m intervals, and analyzed the correlation of the glacial lakes within each elevation range with the temperature and precipitation across the area (Table 5).

We found that there was a significant correlation between the number of glacial lakes and the precipitation at some elevation ranges in the Altai Mountains. Specifically, the elevations of 2.1 to 2.2 km, 2.2 to 2.3 km, and 2.3 to 2.4 km exhibited the correlation coefficients of 0.482, 0.537, and 0.673, respectively. Notably, the terrain within this particular elevation range of the study area is relatively flat. With the rise in precipitation, the number of small glacial lakes sensitive to climate change increased. At elevations ranging from 3.2 to 3.3 km and 3.4 to 3.5 km, a positive correlation was found between the number of glacial lakes and the temperature, with the correlation coefficients of 0.476 and 0.486, respectively. Conversely, within the altitudinal range of 3.3 to 3.4 km, a negative correlation was observed between the area of glacial lakes and the precipitation, with a correlation coefficient of -0.442. Notably, the southern region of the study area, which encompasses most of the

high-elevation areas, has experienced an increase in precipitation and a significant rise in temperature over time. These climatic changes have contributed to the expansion of the glacial lake surface area in the southern region.

**Table 5.** The significant correlation coefficients between glacial lakes and climatic factors at different elevations.

| Elevation (km) | | Count | Area |
|---|---|---|---|
| 2.1–2.2 | Precipitation | 0.482 * | 0.012 |
| | Temperature | −0.351 | 0.010 |
| 2.2–2.3 | Precipitation | 0.537 * | 0.232 |
| | Temperature | −0.283 | 0.104 |
| 2.3–2.4 | Precipitation | 0.673 ** | 0.239 |
| | Temperature | −0.212 | −0.312 |
| 3.2–3.3 | Precipitation | −0.398 | −0.311 |
| | Temperature | 0.476 * | 0.236 |
| 3.3–3.4 | Precipitation | −0.356 | −0.442 * |
| | Temperature | 0.395 | 0.249 |
| 3.4–3.5 | Precipitation | −0.386 | −0.387 |
| | Temperature | 0.486 * | 0.139 |

** $p < 0.01$; * $p < 0.05$.

Previous research has indicated that climatic factors play a pivotal role in driving transformations in lakes located at mid and high elevations. These particular lakes demonstrate a greater susceptibility to temperature fluctuations than that demonstrated by lakes found at lower elevations [66–68]. By analyzing the correlation of glacial lakes at different altitudes with the temperature and precipitation in this area, our results show that there is clear evidence of a positive association between the number of glacial lakes and the temperature in the ranges from 3.2 km to 3.3 km, and from 3.4 km to 3.5 km above sea level. This finding is consistent with previous research on changes in glacial lakes at different elevations in the Tianshan Mountains [66]. It indicates that temperature has a great influence on glacial lake changes at high latitudes and altitudes. When analyzing glacial lake changes, or predicting their evolution, it is necessary to consider how elevation gradients affect the sensitivity of glacial lakes to climate change.

The majority of the water that is found in glacial lakes comes from two sources: glacier meltwater, and precipitation from the atmosphere [69]. If the supply of water is greater than the outflow and evaporation, then the surface area of the glacial lakes increases, and if these conditions are reversed, then it decreases [70]. The temperature in the Altai region has risen sharply since 1996, while precipitation has dropped precipitously. Consequently, during the period of ice and snow accumulation from October to March of the following year, glacial meltwater exerted an important role in the process of glacial lake expansion. Rapid increases in air temperature are the most significant cause of material loss from glaciers [71]. After 1997, the mean annual temperature in the study area started to fall to 0.79 °C in 2000. The number of years that experienced a sharp decrease in precipitation was greater than the number of years with a sharp increase in precipitation from 1990 to 2000, which inhibited the rate of melting of glaciers in that area during that time, and precipitation was unable to meet the needs of the expanding glacial lakes. As a result, there were only 2587 glacial lakes in 2000, covering an area of 582.90 km².

There is a time delay in the response of glaciers to climate change [24]. Rainfall in the study area declined in 2006, while temperature has maintained an increasing trend in the region since 2005. Continued warming has caused a great deal of glacial melting in the north, and the effect of meltwater recharge generated by glacier retreat was larger than the impact of reduced precipitation on the glacial lakes [72]. This accumulation of meltwater may feed the expansion of the glacial lakes for the next couple of years. Between 2006 and 2007, the number of glacial lakes in the northern region increased by 858, while the southern

region experienced a decrease of 572 glacial lakes. The interannual rainfall and temperature fluctuations in the Altai Mountains have slowed since 2014, and the variation trend for the area and the quantity of glacial lakes in the region has gradually stabilized. During 2014 and 2020, the glacial lakes in the Altai Mountains had a mean surface area of 686.86 km$^2$, while the overall number was maintained at 3783.

Considering previous analysis of the temperature trends in and around the Altai Mountains, we believe that the temperature in the region is rising. Thus, glacier retreat in the region has accelerated, and the amount of evaporation has decreased, as the glacial lake continues to expand. In addition, the increased water pressure on the dam body can result in the occurrence of glacial lake outburst floods [73]. Altay is situated in the narrow and long fault valley of the foothills of the lower mountains in the southern Altai Mountains. With drastic changes in the thermal expansion and contraction of the surface, and poor water storage and control functions, the city is the most severely affected by natural disasters, such as flooding and debris flows, due to snowmelt in Xinjiang [74]. Therefore, even though the rate of expansion of the area and the number of glacial lakes in the Altai Mountains will progress slowly, there remains a requirement for a more robust mechanism to gauge and alert for glacial lake flooding in the region. This is necessary to ensure the safety of people's lives, as well as the protection of their property in the area.

## 5. Conclusions

According to the findings of our research, the total surface area of glacial lakes in 2020 was calculated to be 682.38 km$^2$, and the number of glacial lakes was calculated to be 3824. The growth rates of the glacial lake number and area were 47.82% and 17.07%, respectively. The distribution of glacial lakes in the study area showed a larger concentration in the northern region than in the southern region. It was found that glacial meltwater was the main driving force of glacial lake expansion in the Altai Mountains. There are many glaciers in the northern part of the Altai Mountains, which is suitable for the development of new glacial lakes. Further studies have shown that temperature is crucial to the variation in glacial lakes at medium and high altitudes. The high altitude of the southern Altai Mountains, and the significant increase in temperature have created conditions for the rapid expansion of the southern glacial lakes.

Glacial lakes are projected to increase in area and number to 752.70 km$^2$ and 3946 by 2025, and to 793.78 km$^2$ and 4111 by 2030, respectively. In the context of small increases in precipitation and large increases in temperature, glacial lakes with faster area growth rates in the future will mainly be in the southern part of the study area. Against the background of current climate change, the number and area of glacial lakes in the Altai Mountains will maintain a slow growth rate, and the change trend will be consistent with that of glacial lakes between 2014 and 2020. However, glacial lakes in the Altai Mountains exhibit a lagging response to climate change. If temperatures continue to rise, the accumulation of meltwater from glaciers will power the expansion of glacial lakes. This point warrants the attention of policy-making departments.

**Author Contributions:** Data curation, C.M. and Z.L.; writing—original draft preparation, N.W.; supervision, T.Z. and J.Z. All authors have read and agreed to the published version of the manuscript.

**Funding:** This study received funding and support from the Tianshan Cedar Project of Xinjiang Uygur Autonomous Region (No. 2020XS04), and The Third Comprehensive Scientific Investigation in Xinjiang (2021xjkk1001).

**Data Availability Statement:** The data presented are available on request.

**Conflicts of Interest:** The authors declare no conflict of interest.

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
