# Peer review of "Spatio-Temporal Distribution Characteristics of Glacial Lakes in the Altai Mountains with Climate Change from 2000 to 2020"

_remotesensing, doi:10.3390/rs15143689_

Round 1

Reviewer 1 Report

This manuscript (remotesensing-2472423) tries to examine the variations in glacial lake area and number in the Altai Mountains between 2000 and 2020, and to analyze the driving factors behind these changes and combine that result with precipitation and temperature factors based on the MCE-CA-Markov model and Google Earth Engine. Although it fits the aims and scope of this journal, its contribution to glacial lake distribution modeling needs to be explained more clearly. Another concern is that some related latest studies have been neglected. Therefore, a "Major Revision" is required. More detailed comments and suggestions are presented as follows:

- 1. The scientific question or research gap is missing in the Abstract. Similarly, the Introduction Section should also be enhanced because the authors did not raise an important scientific question or gap related to glacier lake evolution studies and beyond this study area. Therefore, potential readers can hardly identify the need that the authors should have to provide a new solution.

- 2. In addition, although a number of previous related studies have been mentioned in the Introduction Section, the limitations of these studies have not been discussed. As a consequence, potential readers will have the feelings that previous related studies have already done enough research in this field (see below for examples).

Monitoring Recent Lake Variations Under Climate Change Around the Altai Mountains Using Multimission Satellite Data. IEEE J. Sel. Top. Appl. Earth Observations Remote Sensing 2021, 14, 599: 1374–1388.

Rapid Expansion of Glacial Lakes Caused by Climate and Glacier Retreat in the Central Himalayas: RAPID EXPANSION OF GLACIAL LAKES IN THE CENTRAL HIMALAYAS. Hydrol. Process. 2015, 29, 859–874.

- 3. Figure 1. Study area and distribution of glaciers and glacial lakes in 2020 and some parts of the introduction of the study areas should be moved to the second section, namely, Materials and Methods.

- 4. In Section 2.1. Data, the data description section did not provide the specific details of all the input data, such as the dates in acquiring them, and spatial resolution. Which year is used for future prediction? I suggest the authors to list all the information in a new table. What is the spatial resolution of the simulated and predicted glacier results? How to deal with all these data with different spatial resolution?

- 5. Did the Landsat images be affected by the cloud? What are the sample data used for measuring the classification accuracies?

- 6. In Section 2.2.4. MCE-CA-Markov: The authors need to explain which driving factors have been selected in this analysis. In addition, what are the years of the driving factors in the MCE-CA-Markov model? Are they consistent with the years of the glacier data?

- 7. Considering that the MCE-CA-Markov model is not the cutting-edge method in this research field. Many latest advanced and cutting-edge land use and land cover change models have not been taken into account in this manuscript, such as: landscape-driven patch-based cellular automaton (LP-CA), and FLUS model (see below).

Modeling urban land-use changes using a landscape-driven patch-based cellular automaton (LP-CA). Cities, 2023, 132: 103906.

Simulating Urban Agglomeration Expansion in Henan Province, China: An Analysis of Driving Mechanisms Using the FLUS Model with Considerations for Urban Interactions and Ecological Constraints. Land 2023, 12, 1189.

- 8. Line 198-199: the authors have mentioned that: "We set the number of cycles to 10 and then used the default Moore 5×5 neighborhood filter to get the simulated image". Why? The authors need to explain clearly the determination of these different parameters.

- 9. In Section 3.1. Glacier lake distribution and changes, the spatial changes of glacier lakes should be displayed based on spatial GIS maps.

- 10. Figure 5. Changes in temperature and precipitation in the Altai Mountains from 1990 to 2020. The R square values are a bit low (0.1448 and 0.2668).

- 11. Figure 9. Prediction of the spatial distribution of land cover in Altai Mountain in the years 2020, 2025 and 2030: the comparison of the land use change simulation results is very difficult to be identified.

- 12. The land use simulation results should be evaluated according to the Figure of Merit (FoM) metric because the calculation of Kappa coefficient does not exclude the non-changed land use pixels. Actually, a null model may even get a better performance if only the Kappa coefficient is considered.

- 13. The authors also need to improve the Conclusion Section by mentioning the main shortages of your work.

- 14. The citation format in the main text is incorrect. For example, only the surname of the authors is needed.

Extensive editing of English language required.

Author Response

Dear reviewer,

We are very grateful to your comments for the manuscript.  According with your advice, we tried to amend the relevant part and made some changes in the manuscript. All of your questions were answered below. Please see the attachment. 

We appreciate for your warm work earnestly, and hope that the correction will meet with approval.  Should you have any questions, please contact us without hesitate.  Once again, thank you very much for your comments and suggestions. 

Yours Sincerely, 

Nan Wang

Reviewer 2 Report

The spatial and temporal variations of glacial lakes in the Altai were analyzed with Google Earth engine in this article. The Markov model was used to predict the development of glacial lakes in the Altai Mountains. It is shown that since 2014, interannual fluctuations in precipitation and temperature in the Altai Mountains have slowed down, and the trend in the area and number of glacial lakes in the region has reached a point of gradual stabilization. Although the area and number of glacial lakes will continue to increase over the next decade, the pace of development is expected to slow down.

The introduction describes the problem of glaciers that melt and glacial lakes form. It is proposed to evaluate the dynamics of these glacial lakes. However, in the introduction there is no information on remote sensing of glacial lakes in the world. This is a significant drawback of the introduction. The subject of the journal primarily involves the use of remote sensing methods, therefore, the introduction should be redone with an emphasis on remote sensing of glaciers and glacial lakes. It is also possible to specify about the methods that will be used in this article, for example, about using GEE. Accordingly, the list of references should be redone with an emphasis on the above.

Author Response

(The authors gave the same response as above.)

Round 2

Reviewer 1 Report

This is the second time I am reviewing this manuscript. I would like to remind the authors that the "Remote Sensing" is a considerably prestigious journal and thus very competitive to publish. The scientific rationale behind the proposed research framework aimed at publishing such a prestigious journal should be crystal clear. I believe, authors could have taken the opportunity of the first revision and significantly improve their work based on the reviewer's comments and suggestions. Unfortunately, there are still major concerns remaining in this manuscript, which I believe they are either not explained well or they are somehow misleading. My comments and suggestions are summarized below:

- 1. The scientific gap is still missing. Expand on the context and rationale for studying the relationship between the area, quantity and distribution of glacial lakes in Altay Mountain. Why is it important to investigate the relationship in this specific study area? What existing gaps in knowledge does this research aim to fill? Note that such relationship has already been analyzed in many other study areas.

- 2. In Section 2.1. Data, I am still missing the acquire dates in Table 1. Landsat images applied for this study.

- 3. The authors still failed to explain why they only selected those driving factors in this analysis. What about the factors related to the man-made environment? In addition, this manuscript did not consider the serious problem of the multi-collinearity of different driving factors.

- 4. The authors claimed that since their module only has a single Kappa coefficient test, so they merely used the Kappa for evaluating the land use simulation results, which is totally unreasonable. Again, a null model may even get a better performance if only the Kappa coefficient is considered.

- 5. A critical issue to be addressed in the new version is that how the policy-makers can validate the authors’ proposal. I mean, how can the authors validate the data predictions to the future if we do not know how the nature or society will behave?

- 7. Basically, there is no real discussion. How to link the results and key findings in this paper with the previous findings and conclusions? The novelty/originality should be clearly justified that the manuscript contains sufficient contributions to the new body of knowledge from an International perspective. There are already many academic studies about this old topic. What new global knowledge can this paper contribute to the existing international literature?

- 8. The scale effect is also worth discussing. What is the difference between the simulation results at different resolutions? What are its impacts on accuracy?

- 9. In particular, the Result Section should be enhanced to compare this study with the results from other recent advances. The analysis of the results lacks depth and only stays at the level of textual description.

- 10. Some grammatical errors, typos can still be found in the manuscript. A proof reading by a native English speaker should be conducted to improve both language and organization quality.

- 11. Conclusions Section: in this part, the conclusions just simply repeated the results. In addition, what are the advantages and disadvantages of the model/results compared with previous similar studies? These sentences are not conclusions. In other words, the manuscript needs a strong take-home message in order to prove its value.

Some grammatical errors, typos can still be found in the manuscript. A proof reading by a native English speaker should be conducted to improve both language and organization quality.

Author Response

Dear reviewer,

We are very grateful to your comments for the manuscript again.  According with your advice, we tried to amend the relevant part and made a second revision to our manuscript. All of your questions were answered below. Please see the attachment. 

We appreciate for your warm work earnestly, and hope that the correction will meet with approval.  Should you have any questions, please contact us without hesitate.  Once again, thank you very much for your comments and suggestions. 

Yours Sincerely, 

Nan Wang
